# The Regulation of NFE2L2 (NRF2) Signalling and Epithelial-to-Mesenchymal Transition in Age-Related Macular Degeneration Pathology

**DOI:** 10.3390/ijms20225800

**Published:** 2019-11-18

**Authors:** Juha M. T. Hyttinen, Ram Kannan, Szabolcs Felszeghy, Minna Niittykoski, Antero Salminen, Kai Kaarniranta

**Affiliations:** 1Department of Ophthalmology, Institute of Clinical Medicine, University of Eastern Finland, P.O. Box 1627, FI-70211 Kuopio, Finland; Kai.Kaarniranta@uef.fi; 2The Stephen J. Ryan Initiative for Macular Research (RIMR), Doheny Eye Institute, DVRC 203, 1355 San Pablo Street, Los Angeles, CA 90033, USA; 3Institute of Biomedicine, University of Eastern Finland, P.O. Box 1627, FI-70211 Kuopio, Finland; Szabolcs.Felszeghy@uef.fi; 4Institute of Dentistry, University of Eastern Finland, P.O. Box 1627, FI-70211 Kuopio, Finland; 5A. I. Virtanen Institute for Molecular Sciences, University of Eastern Finland, P.O. Box 1627, FI-70211 Kuopio, Finland; Minna.Niittykoski@uef.fi; 6Department of Neurology, Institute of Clinical Medicine, University of Eastern Finland, P.O. Box 1627, FI-70211 Kuopio, Finland; Antero.Salminen@uef.fi; 7Department of Ophthalmology, Kuopio University Hospital, P.O. Box 100, 70029 KYS Kuopio, Finland

**Keywords:** age-related macular degeneration, epithelial-to-mesenchymal transition, hypoxia, nuclear factor erythroid 2-related factor, oxidative stress, reactive oxygen species, retinal pigment epithelium

## Abstract

Age-related macular degeneration (AMD) is a mounting cause of loss of sight in the elderly in the developed countries, a trend enhanced by the continual ageing of the population. AMD is a multifactorial and only partly understood, malady. Unfortunately, there is no effective treatment for most AMD patients. It is known that oxidative stress (OS) damages the retinal pigment epithelium (RPE) and contributes to the progression of AMD. We review here the potential importance of two OS-related cellular systems in relation to AMD. First, the nuclear factor erythroid 2-related factor 2 (NFE2L2; NRF2)-mediated OS response signalling pathway is important in the prevention of oxidative damage and a failure of this system could be critical in the development of AMD. Second, epithelial-to-mesenchymal transition (EMT) represents a change in the cellular phenotype, which ultimately leads to the fibrosis encountered in RPE, a characteristic of AMD. Many of the pathways triggering EMT are promoted by OS. The possible interconnections between these two signalling routes are discussed here. From a broader perspective, the control of NFE2L2 and EMT as ways of preventing OS-derived cellular damage could be potentially valuable in the therapy of AMD.

## 1. Introduction

### 1.1. Age-Related Macular Degeneration (AMD)

Age-related macular degeneration (AMD) is the principal cause of visual loss in the developed countries and, since its incidence increases steadily with age, the affected population shows a continual increase [1]. Approximately 23% of individuals between 65 and 74 years display some signs of AMD and this percentage rises to 35–40% in those individuals between 75 and 84 years [2]. It is estimated that the numbers of AMD cases will double in the next 40 years among people ≥60 years old due to the overall increase in the aged population [2,3,4]. According to current understanding, AMD develops in a multifactorial manner. In addition to ageing, the main risk factor for AMD is family history. Smoking, high-fat diet, obesity, hypercholesterolemia and hypertension are linked to AMD development [5,6]. With respect to the genetic risks, variations in complement factors (H, C3 and I), age-related maculopathy susceptibility 2 and clusterin-like protein loci are thought to be of importance [5,6,7]. One early concept in ophthalmology, the proposal that sunlight/UV exposure to the eye would be a direct major risk for AMD, has however recently been criticized as controversial results have been published [8,9].

The crucial aspect of this disease is the damage in the central retina or macula (Figure 1). AMD is classified into two main forms. The more abundant (about 85–90% of cases) non-exudative dry form progresses slowly [10]. Currently, there is no effective therapy to dry AMD. The dry form of AMD is usually a risk factor for or precursor to, the more severe exudative, wet form. This accounts for much fewer (10–15%) of all AMD cases. In wet AMD reduced central vision emerges rapidly over days and weeks. The wet form of AMD is associated with neovascularisation, when choroidal vessels penetrate into Bruch’s membrane (BM) and retinal pigment epithelium (RPE) in the macula. Ultimately, the accumulation of sub-retinal fluid occurs, leading to macular oedema (a characteristic hallmark of wet AMD), haemorrhage and fibrosis (Figure 1C,E) resulting in a permanent impairment of vision [11]. Currently, the wet AMD is rather effectively treated with anti-vascular endothelial growth factor (VEGF) injections [12].

The retinal pigment epithelium (RPE) cells have an important role in the pathology of AMD. RPE is a one-cell layer located between photoreceptors at the front and BM and choriocapillaris at the back (Figure 1D,E). Not only the RPE but also the photoreceptor outer segments (POS) and the choroid, are degraded in AMD. The RPE has the responsibility of being both the guardian and caretaker of the photoreceptors. It transports nutrients from the capillary layer to the photoreceptors, is involved in the recharging of the visual cycle, that is, the conversion of all-*trans* retinol to 11-*cis*-retinal and undertakes the phagocytosis of waste material continuously budding from POS. It produces and secretes several growth factors such as pigment-epithelium-derived factor and VEGF. Furthermore, the RPE absorbs light, especially UVA radiation, because it contains pigments (like melanin, flavins and retinoids) and in this way it helps to prevent phototoxicity. These compounds are especially abundant in the macula and are seen as a yellowish pigmentation. Disturbances in these processes lead to the degeneration of the retina and eventually to the loss of vision [7,13,14,15].

The visible hallmarks of AMD are insoluble aggregation deposits—called drusen—between BM and RPE. Drusen consists mainly of lipids, including cholesterol, calcium phosphate and several proteins [16]. The drusen types, sizes and numbers are predictors of the progression of AMD [17]. Moreover, the thickening of the BM, a phenomenon connected to fibrosis, is observed in AMD [18,19,20]. These changes may be responsible for the disruption of the transfer of nutrients and macromolecules from the choroid to the RPE [21]. In the advanced stage of AMD, a loss of choriocapillaris is observed [22].

Lipofuscin is another type of aggregation; it is a pigment which accumulates within RPE cells progressively with ageing. It is comprised mainly of lipids and proteins and produces the highest autofluorescence intensity in RPE and this can be used as a biomarker in ophthalmological clinics [23]. At present, although the role of lipofuscin in AMD is unclear [24], it is thought to be an indicator of RPE stress and a potential sign of AMD and not simply an age-related pigment [25].

### 1.2. Retinal Pigment Epithelium (RPE) and Oxidative Stress

The exposure of the retina to reactive oxygen species (ROS) is thought to be a crucial factor in the development of AMD. The RPE is prone to high oxygen tension due to its close proximity to the choriocapillaris. In fact, it is considered to be exposed to the most intensive O_2_ pressure of any human tissue [26]. In addition, RPE has a large number of mitochondria, due to its high energy requirements and elevated metabolic rate. This leads to the production of high levels of ROS, the by-products of oxidative phosphorylation, in the RPE [26]. Roehlecke et al. [27] have shown that under blue light exposure, intense oxidative stress (OS) occurs in the POS, the region which RPE “cleanses” (including the removal of the end-products of the visual cycle) and this forms an important source of the waste accumulated into the RPE. 

As the individual grows older, the functional activity of the RPE declines and its capacity to neutralize ROS is reduced. The endoplasmic reticulum stress, unfolded protein response, causing protein aggregation, increased DNA damage due to light and ROS exposure increase the signs of ageing [28,29]. The accumulation of damaged biomolecules places a larger workload on the cell. As these processes (such as chaperone function, proteasomal clearance, lysosomal activity and autophagy and its subtypes) also have reduced activity. This leads to the accumulation of potentially toxic deposits (drusen and lipofuscin) that can further damage the RPE and impair its functional activity. Toxic materials, for example the autofluorescent lipid-protein aggregates, contribute further to ROS production [18,30,31,32]. OS present in the retina causes a chronic inflammation response, which has been linked to disturbances in the functions of the RPE and thus to the progression of this disease. For example, nucleotide-binding oligomerisation domain-like receptor family, pyrin domain-containing-3 (NRLP3) inflammasome receptor is activated in RPE, induced by drusen and lipofuscin components [33].

Melanin granules in RPE and choroidal melanocyte cells also have a role in ROS balance since they absorb light and thus possess a protective role against UV radiation at the base of the retina. It is possible that melanin removes ROS or inhibits their formation [34]. In contrast, it has been claimed that when melanin is irradiated by UV, it produces ROS [35,36,37]. This might represent another ROS burden on RPE, antagonizing the beneficial effect of melanin to combat against ROS. As the number of melanosomes decreases with ageing, however, there is a decrease in the melanin content in RPE cells in AMD [38].

In addition, another type of OS, hypoxia might promote AMD. As the macula receives the greatest blood flow and the highest oxygen supply than the other tissues, any reduction in the blood supply may evoke hypoxia in the macula. The dry and wet forms of AMD also cause a reduction in choroidal perfusion [39]. Therefore, hypoxia be might be either a cause or consequence of the disease

In this review we discuss two systems connected to OS. The first is the Kelch-like ECH-associated protein 1–nuclear factor erythroid 2-related factor 2 (KEAP1-NFE2L2) pathway (Section 2), an important route of sensing and responding to the presence of electrophiles and OS. Activation of this pathway leads to the expression of antioxidant genes, which are crucial in the neutralization of the disturbed redox balance. Knock-out animal models of the *NFE2L2* gene have been established. In the eyes of these animals, many signs of disturbed clearance, accumulated deposits, RPE degeneration and, finally, age-related visual loss are seen. These findings link the damage to *NFE2L2* gene deficiency in the pathogenesis of AMD [40,41]. These models are discussed in more detail in Section 2.6.

The second pathway, which is also linked to OS, is the epithelial-to-mesenchymal transition (EMT) (Section 3), a shift in the cellular phenotype, that is, the cell changes from being polarised, layered and immobile to a non-layered, spindle-like cell which is capable of migration. Both of these systems are thought to be connected with cellular changes and damage in the retina and thus have been linked with age-related diseases, such as AMD. The type 2 of EMT (Section 3.1), which is associated with fibrosis, following choroidal neovascularisation, wound healing and scar formation, is of importance in the final stage of AMD, leading to blindness [42]. In addition, the known interconnections between KEAP1-NFE2L2 pathway and EMT will be examined. 

## 2. The Kelch-Like ECH-Associated Protein 1 (KEAP1)-Nuclear Factor Erythroid 2-Rlated Factor (NFE2L2) Pathway

### 2.1. Overview

The key route for sensing and responding to the OS response in the cell is primarily attributable to the function of two proteins—Kelch-like ECH-associated protein 1 (KEAP1) (Section 2.3) and nuclear factor erythroid 2-related factor 2 (NFE2L2) (Section 2.2). The former acts as a stress sensor and is the master regulator of the oxidative and electrophilic stress. Receptor protein NFE2L2 in turn transmits the stress signal to the nucleus, where it acts as a transcription factor to activate the coordinated system of antioxidant and anti-inflammatory responses. This signalling route is referred to as the KEAP1-NFE2L2 pathway.

Before discussing this pathway, we must stress that there is considerable confusion around the terminology, a topic already addressed by Baldelli et al. [43]. NFE2L2 has often been referred to in the literature as NRF2 or Nrf2. This has caused some confusion, since another protein—nuclear respiratory factor 2—has also been abbreviated as NRF2. The latter protein, also known as GA-binding protein transcription factor α (GABPA), a multimeric transcription factor, might even have some overlap with the functions of NFE2L2 [43], which will not be addressed in this review. Hence for clarity, we shall use consistently the abbreviation of NFE2L2 for the protein.

NFE2L2 is an essential key effector in the sensing of endogenous and environmental oxidative and electrophilic stress in the cell. NFE2L2 is a transcription factor which activates signalling pathways that ensure the maintenance of the cellular redox state by activating phase II detoxification and antioxidant enzymes [44,45,46,47,48]. In addition, NFE2L2 participates in the quality control of proteins that involves the removal of misfolded and ubiquitinated proteins and the clearance of damaged cellular organelles. It activates the expression of subunits of the proteasomal machinery and the autophagy-related genes [49] (Section 2.5). It is believed that the KEAP1-NFE2L2 pathway induction and subsequent stimulation of the antioxidant response could prevent or slow the progression of tumour growth, vascular diseases and degenerative and immunological diseases, both in early stages and in subsequent interventional therapy. Similarly, ageing and senescence could be counteracted by augmenting the activity of NFE2L2, thus promoting longevity [49,50,51,52]. In Section 2.2, Section 2.3, Section 2.4, we will first discuss the canonical mechanism of the KEAP1-NFE2L2 antioxidant response pathway. 

### 2.2. NFE2L2, the Master Regulator of Oxidative Stress Response

NFE2L2 belongs to the family of cap’n’collar (CNC) proteins, whose members contain a basic leucine-zipper domain (bZIP). The family consists of six known members subdivided into two groups. The first group comprises four nuclear-factor erythroid-derived factors: p45 NF-E2 and NFE2L1, 2 and 3, while the second two Broad-complex, Tramtrac and Bric-à-brac (BTB) factors: BTB and CNC homolog proteins 1 and 2 (BACH1 and 2) [53,54,55]. Depending on the ability of CNC proteins to bind to a transcription co-factor, they either stimulate or arrest the transcription of antioxidant genes. 

NFE2L2 has seven NFE2L2 (NRF2)-ECH homology domains (Neh1-7). Neh1 contains the leucine zipper, which is responsible for the DNA binding and dimerization with its cofactors and regulators, like small musculoaponeurotic fibrosarcoma proteins (sMAFs) and Jun proteins. This heterodimerisation is a pre-requisite for the efficient binding to the antioxidant response elements (AREs) in DNA (Section 2.4). The N-terminal Neh2 domain is considered to be the site which binds to KEAP1. There are two motifs, ^29^DLG (Asp-Lys-Gly) and ^79^ETGE (Glu-Thr-Gly-Glu), involved in this mechanism. The Lys residues between these motifs are subjected to ubiquitination and this promotes the rapid turnover of NFE2L2 by proteasomal degradation [56]. Domains Neh3-5 are responsible for the transactivation of NFE2L2. For example, Neh4 and 5 interact with the nuclear co-regulatory receptor-associated cofactor 3-amplified in breast cancer 1 (AIB1)-steroid receptor coactivator-3 (RAC3-AIB1-src-3), enhancing the activity of the NFE2L2 in the expression of the ARE-dependent genes [57]. In turn, domains Neh6 and Neh7 provide the post-translational regulation of NFE2L2. For example, retinoic X receptor α interacts with Neh7 domain, leading to the repression of NFE2L2 [58]. Human NFE2L2 also contains six cysteines, which are susceptible to oxidative modification. In addition, the phosphorylation of Ser and Thr residues and the acetylation of Lys residues are important in the transcriptional activity of NFE2L2 [59,60]. For example, the activity of NFE2L2 may be altered at the levels of transcription, translation, post-translational modifications, nuclear translocation and the binding to the promoters of regulated genes [61]. 

The KEAP1-NFE2L2 pathway can be activated by a wide range of chemical compounds having oxidative or electrophilic activity, as well as non-electrophilic agents, which can be of either an exogenous or endogenous origin. These include toxicants and cancer-chemopreventive agents that include polyphenols, organosulfurs, isothiocyanates, heavy metal salts, plant-derived compounds and many synthetic herbicides like paraquat. Examples of endogenous agents include hydrogen peroxide, nitrogen oxide, fumarate and several lipid peroxidation products (reviewed by Sihvola and Levonen [47]). Many of these agents are produced in the RPE and are thought to promote AMD [62]. Furthermore, NFE2L2 is also activated by lifestyle factors such as tobacco smoking. In fact, smoking is considered to be an important environmental factor for AMD (Section 1.1), since many of the compounds present in tobacco smoke, for example, acrolein, cadmium and hydroquinone, cause excessive oxidative damage [63]. To combat their detrimental effects, the KEAP1-NFE2L2 pathway has a prominent role in mounting the anti-oxidant defence of the cells in the retina [64].

According to current knowledge, NFE2L2 directly regulates about 250 genes involved in cellular homeostasis. These include antioxidant proteins, detoxificants, drug transporters and other cytoprotective proteins, for example NAD(P)H:quinone-oxidoreductase 1 (NQO-1), γ-glutamylcystenyl synthetase, glutathione-S-transferase and heme oxygenase 1 (HO-1) [65]. Perturbations in their levels could be associated with AMD pathogenesis and conversely their activation could be beneficial in combatting this disease [66,67,68,69]. NFE2L2 and related factors act as primary regulatory molecules, that is, they recognize the presence of a threat and bind to AREs (Section 2.4), called in a broader sense as electrophile response elements [70,71]. NFE2L2 has several partners in DNA recognition, that is, members of activator protein-1 family, including activating transcription factor 4 (ATF4), sMAFs, like MAFF, MAFG and MAFK and JUN transcription factors like c-JUN, JUN-B and JUN-D [70,72,73]. 

### 2.3. Regulation of NFE2L2 by KEAP1, the Sensor of Oxidative Stress

KEAP1 is a cytoplasmic protein homologous to the *Drosophila* actin-binding protein [74,75]. It has two important binding domains—the BTB and Kelch-domain (double glycine-repeat), flanked by N- and C-termini and an intervening region (IVR) in the middle. KEAP1 exists in a dimeric form in the oxidation sensing complex with the BTB region being responsible for the dimerization. The IVR domain of KEAP1 binds to an adaptor component, Cullin 3 (CUL3). This becomes NEDD8ylated (i.e. modified by the binding of a small ubiquitin-like molecule) [76]. In addition, this complex contains RING box protein 1 (RBX1) and this subsequently binds to an ubiquitin E3 ligase [74,76,77]. KEAP1 is a thiol-rich protein—the human isoform contains 27 cysteine residues [78]. 

The regulatory KEAP1-NFE2L2 binding is described as a “hinge-and-latch” mechanism. The two Kelch domains of KEAP1 dimer partners bind to two positions in the Neh2 domain of NFE2L2: The weaker (“latch,” closed conformation) bond is formed by the ^29^DLG motif and the higher-affinity (“hinge”, open conformation) by the neighbouring ^79^ETGE motif (Figure 2A). When both of these binding sites are occupied, the now immobilised NFE2L2 is constitutively ubiquitinated by the E3 ubiquitin ligase in the KEAP1 complex [56,79,80]. This modification occurs at seven Lys residues which are situated between the KEAP1 binding sites of NFE2L2 and it leads to the rapid degradation of NFE2L2 by the 26S proteasome machinery [75,81]. The release of NFE2L2 and its transfer to the proteasome is mediated by a complex comprising valosin-containing protein (VCP/p97), UBA-UBX containing protein UBNX7 and Ub-binding heterodimeric cofactors NPL4 and UFD1. This complex can become attached to both NEDD8ylated CUL3 and ubiquitinated NFE2L2 [82]. 

The key event in the launching of the stress response via NFE2L2 is the attachment of electrophilic ligands to the Cys residues of the sensor protein KEAP1, for example, due to redox stress (Figure 2B). Alternatively, non-electrophilic inducers of NFE2L2, which are less ubiquitous, hinder the DLG binding by blocking the binding to the Kelch domains in the KEAP1 complex. These inducers most likely disrupt the DLG binding because their binding affinities are weaker than that of the ETGE motif itself [48]. Although there are 27 Cys residues in the human KEAP1, the most reactive and most important as oxidation sensors have been found to be Cys-151, 257, 273, 288 and 297 and 433 [47,83,84,85]. The first of these cysteines is located in the BTB domain and the last in the Kelch domain, while the others are placed in the intervening region. The first cysteine seems to be the most important as an NFE2L2 inducer, although it is not located near to the “latch” binding site. Instead, it might interfere with the connection between KEAP1 and CUL3 [86]. 

Amino acid replacement studies have revealed that these Cys residues can act specifically and/or collaboratively as sensors and thus are capable of detecting a broad range of environmental stresses [84]. For example, nitric oxide modifies Cys-151, 4-hydroxynonenal residues home in on Cys residues 273 and 288 and Cys residue 434 senses the exposure to 8-nitro-guanoside cyclic monophosphate [47]. When one examines the situation in AMD, it has been postulated that dimethylfumarate, a protector of RPE cells from oxidative injury [87], can be sensed by the Cys-151 modification, whereas cadmium, a potential promoter of AMD which is present in cigarette smoke, is a Cys-151, 273 and 288-independent activator of KEAP1-NFE2L2 pathway [84,88]. 

The KEAP1-NFE2L2 pathway also interacts with other antioxidant pathways. For example, one of the best characterised is its interaction with silent information regulator 2-related proteins (sirtuins), a family of NAD+-dependent histone deacetylases [89]. The sirtuins modulate many cellular processes associated with antioxidant and redox signalling. For example, during OS, sirtuin 1 deacetylates NFE2L2, leading to reduced ubiquitination and subsequently to increased activity of KEAP1-NFE2L2 signalling [90]. 

### 2.4. The Transmission and the Action of Oxidative Stress Response in the Nucleus

The mechanism for NFE2L2 activation as a consequence of the oxidation of the Cys residues involves the loosening of the “latch” DLG-bond. Subsequently, NFE2L2 is still bound to KEAP1 via the higher affinity “hinge” ETGE attachment but this slackening of its three-dimensional structure is sufficient to allow the termination of ubiquitination by the E3 ubiquitin ligase bound to the KEAP1-CUL3-RBX1-complex (Figure 2A) (Section 2.3). NFE2L2 is still tied to KEAP1 by the ”hinge” bond and is modified, before the final release from the complex. These modifications include phosphorylations by atypical protein kinase Cs (aPKCs), for example, to Ser-40 by PKCδ and PKCι, Ser-558 phosphorylation by AMPK, acetylation induced by cAMP response element-binding protein (CREB) (Figure 2B) and others at present unknown phosphorylation reactions [60,91,92,93]. 

The nuclear translocation of NFE2L2 is assisted by karyopherins α1 and β1 through the nuclear pore complex, aided by the nuclear localisation sequence (NLS) in the N-terminal Neh2 domain of NFE2L2 [94,95]. The modification of Cys-183, which is located in the transactivation domain (Neh5) of NFE2L2, inactivates the nuclear export signalling activity [96]. When it is located in the nucleus, NFE2L2 forms a heterodimer with sMAFs, like MAFF and MAFK [97]. Furthermore, heterodimerisation with other transcription factors, like c-JUN and JDP52, increases the transcriptional activity. In addition, further acetylation and SUMOylation of NFE2L2 are needed for its ARE binding [98,99]. Hydroquinone, an electrophile which has been linked to the development of AMD, prevents the association of BACH1 with NFE2L2 and thus activates NFE2L2-mediated oxidation defence [100]. The dimer of ATF4 and NFE2L2 augments the expression of HO-1 [73]. There are several factors that negatively regulate NFE2L2’s activity. For example, c-MYC competes for binding to secondary coactivators and BACH1 and 2, the family members of NFE2L2, form heterodimers with sMAFs and therefore hinder the NFE2L2-ARE contact [54,101]. It has been recently reported that the levels of BACH and c-MYC increase with ageing and this might lead to the decrease of NFE2L2-ARE signalling [102]. 

Finally, NFE2L2 acts as a transcription factor triggering the expression of ARE promoter-dependent genes. The AREs are *cis*-acting promoter elements of genes and have a common consensus sequence, that is, 5’ RGTGACnnnGC [103,104]. They are located about 80 to 750 bases upstream of the transcription initiation site [105,106]. AREs are connected to genes expressing antioxidant proteins, phase II detoxification enzymes, drug transporters and other cytoprotective proteins [44]. Phase II enzymes conjugate attenuators (such as reduced glutathione, amino acids and other sulphur containing compounds) to toxic electrophilic compounds, for example, the AMD-related and cigarette smoke-derived antioxidants hydroquinone and acrolein [63,107], thus reducing their detrimental effects. Mutation studies have revealed that Cys-119, 235 and 506 are required for the binding of NFE2L2 to the AREs and to the co-activator or secondary co-activator proteins, for example, p160 nuclear receptor coactivator proteins, which in turn bind to tertiary regulators such as CREB-binding protein-binding protein and its paralog p300 protein [108].

NFE2L2 is thought to remodel chromatin structures and facilitate the formation of the general transcription machinery via its secondary and tertiary coactivators [71]. CREB even directly augments the binding of NFE2L2 to AREs, by acetylating Lys-596 and 599 [60,99]. This multilayered machinery represents a higher order for the regulation of transcription. It has been found that in addition to the stimulus evoked by inducer, the time that NFE2L2 remains active is dependent on the age of the animals. *tert*-Butylquinone promotes ARE binding for up to 30 min in primary astrocytes from old rats, whereas this lasts up to 3 h in cells from young animals [109]. This might be of importance in relation to how well ageing cells can combat OS and therefore it may be involved in the development of AMD in the eyes of older people.

Subsequently, other proteins in the transcription “preinitiation complex” are recruited, though the order is still unclear. CREB binds to the Neh3 and Neh5 domains of NFE2L2 and then acetylates the lysine residues of the Neh1 and Neh3 domains and possibly also the surrounding histones, like Lys 27 of histone 3, located in chromatin around ARE sequences. These are changes that increase ARE-driven expression [99,110]. Moreover, the histone acetyltransferase, MOF, may be necessary, as it acetylates NFE2L2 and its knock-down has been found to decrease ARE-regulated gene expression [111]. Finally, the other moderators of NFE2L2-mediated transcription include mediator (MED) subunits 1, 16, 23 and 24, ATF4, RAC3 and chromodomain helicase DNA binding protein-6 (CDH6) [57,112]. The whole complex transmits its signal initially detected by KEAP1 to RNA polymerase II and the expressions of OS-induced antioxidant and metabolic genes are eventually [110]. When the basal conditions after OS have been restored, KEAP1 continues to regulate the activity of NFE2L2. KEAP1 is translocated to the nucleus, where it binds to NFE2L2 leading to its dissociation from the ARE. The complex is then transported to the cytoplasm where it associates with the CUL3-RBX1-E2 ubiquitin ligase complex, resulting in the termination of the antioxidant signalling by NFE2L2, as described previously in this section [113]. 

NFE2L2 is upregulated in some longevity-supporting interactions, like caloric restriction and the NFE2L2/KEAP1 pathway might be crucial in the determination of species longevity [114]. The total expression level of NFE2L2 might be higher in long-living species as that of KEAP1 is lower. Furthermore, the nuclear binding activity of NFE2L2 is higher in these species as are the transcription levels of its downstream targets, the anti-oxidation [49]. 

The KEAP1-NFE2L2 pathway not only exerts beneficial effects on cellular balance, it may also display a detrimental side. While the activation of NFE2L2 has been evaluated in cancer prevention in clinical trials, it has been claimed that the constitutive activation of NFE2L2 contributes to the survival and growth of cancer cells in many types of tumours [115]. NFE2L2 has been shown to support the progression of several types of cancers and to promote their resistance to chemo- and radiotherapy [116,117,118]. There is a report that the survival of certain cancer cells can occur when mutations cause a permanent activation of this stress-related pathway [119]. 

#### 2.4.1. Non-Canonical KEAP1-NFE2L2 Regulation

In addition to the canonical KEAP1-NFE2L2 sensing of OS discussed above, there are non-canonical mechanisms, mediated by other proteins, which can bind to either KEAP1 or NFE2L2 and thus compete with their reciprocal binding to regulatory sites [94,120]. The scaffold protein, sequestosome 1/p62 (SQSTM1/p62) (discussed in connection with autophagy in more detail in Section 2.5), the protein partner and localiser of BRCA2 (BALB2; BRCA2, Breast cancer type 2 susceptibility protein), dipeptidyl peptidase III (DPP3), Wilms tumour gene on X-chromosome (WTX) and prothymosin α can all directly interact with KEAP1. On the other hand, p21 (p21^CIP1/WAF1^), a cyclin-dependent kinase inhibitor, can interact with the DLG and ETGE sites of NFE2L2 [121]. The interactions with KEAP1 are mediated by motifs identical to or resembling the ^79^ETGE, which is essential for the KEAP1-NFE2L2 interaction (Section 2.3). For example, this can occur in the case of SQSTM1/p62 via a ^350^DPSTGE motif in its KIR-domain. All of these interactions can disrupt the KEAP1-NFE2L2 complex, thus releasing NFE2L2 to the nucleus and consequently the expression of ARE-promoted genes [122,123,124,125]. 

A specific inhibitor of p70 ribosomal kinase S6 kinase, PF-4708671 induces apoptotic cell death. Park and others [126] have shown that PF-4708671 induces autophagic KEAP1 degradation-mediated activation of NFE2L2 via a SQSTM1/p62-dependent mechanism. Thus, the SQSTM1/p62-dependent NFE2L2 activation was able to protect cells from apoptosis induced by this kinase inhibitor which suggests that NFE2L2 is involved in autophagy via the action of SQSTM1/p62. This could also be a plausible way to prevent AMD prevention via autophagy induction.

p21 (p21^CIP1/WAF1^) is an important regulator of many cellular processes, for example, cell cycle regulation, DNA replication, cell differentiation, apoptosis and it is upregulated in response to OS in order to promote cell survival [127]. An NFE2L2 stabilisation process has been revealed, where p21 competes with KEAP1 for DLG “latch” binding to NFE2L2 [121]. It has been reported that p21 is upregulated in ARPE-19 cells after exposure to cigarette smoke concentrate; this was accompanied by increases in the levels of ROS and numbers of DNA lesions, as well as activation of senescence [128]. As p21 can also inactivate the KEAP1-NFE2L2 pathway, due to its ability to bind to both of these proteins, one could assume that any increase in its levels might influence the AMD progression via the production of excessive amounts of ROS. This adds another dimension to the complexity of the NFE2L2-mediated response against OS. 

In addition to the KEAP1-regulated ubiquitination of NFE2L2, there exists a redox-independent β-transducin repeat-containing protein (β-TrCP)-mediated ubiquitination pathway. This protein acts as a receptor for the Cullin 1 (Cul1)-RBX1-containing ubiquitin ligase complex. Two β-TrCP binding motifs exist in the Neh6 domain of NFE2L2. Glycogen synthase kinase 3β (GSK-3β) phosphorylates the first of the motifs in Neh6, which allows the protein to bind to β-TrCP and further to Cul1 and the ubiquitination machinery. There are external signals (like cytokines, growth factors and neuromediators) that inhibit GSK-3β and thus the ubiquitination of NFE2L2 is terminated and this transcription factor is released [129]. Ebrahimi et al. [130] have shown that if mice are exposed to either a high fat diet or cigarette smoke, there is an increase in the level of GSK-3β in the RPE of those animals’. This is followed by a decrease in the levels of NFE2L2 and a weakened oxidative response mechanism that may also be involved in the disease phenotype of AMD.

#### 2.4.2. KEAP1-Independent Control of NFE2L2

One KEAP1-independent mechanism to regulate NFE2L2 involves the nuclear receptor-mediated control of NFE2L2 signalling [115]. There are several factors that can act in this manner, for example, activating transcription factor 3 (ATF3) [131], p53 [132], oestrogen-receptor β (ERRβ) [133] and glucocorticoid receptor (GR) [134]. Furthermore, these factors could also include nuclear factor kappa-light chain enhancers of activated B cells (NF-κB) [135] and the participation of retinoid receptors in the inhibition of NFE2L2. The role retinoids will be discussed in more detail in Section 2.6 In addition to the above factors, tumour suppressor protein p53 induces cell cycle arrest and apoptosis. The accumulation of ROS is required to trigger apoptosis and this phenomenon comes into play when NFE2L2 is downregulated and consequently NQO-1 and glutathione-S-transferase are suppressed.

The nuclear factor kappa-light chain-enhancers of activated B cells (NF-κBs) belong to a family of transcription factors regulating many genes and their actions result in a broad variety of responses, depending on the cell type and stimuli being studied [136]. The p50-p65 dimer of NF-κB, which is the major form of NF-κB present in most cells, operates as a counterweight to NFE2L2 in the nucleus with respect to the expression of the genes involved in inflammatory, immune and acute phase responses [135,137]. When both NFE2L2 and NF-κB protein levels are simultaneously increased in the nucleus, they compete for binding to CREB. Transcription factor CREB is indispensable for the activity of both of these proteins. Moreover, NF-κB recruits histone deacetylase 3 (HDAC3) and the resulting insular histone hypoacetylation decreases the transcriptional activity of NFE2L2. Inhibition of NF-κB by suppressors of the inflammatory response leads to NFE2L2 activation. In addition, the suppression NF-κB activity triggers the conversion inside the cell to a more reductive milieu [135,138,139]. The prevention NF-κB signalling achieved by the activation of NFE2L2 might be advantageous in the prevention of AMD, since the augmented chronic inflammatory response has been linked to disturbances in the functions of the RPE and thus to the progression of this disease [140,141]. 

### 2.5. NFE2L2 Signalling and Autophagy

Macroautophagy, later referred to simply as autophagy, is an important cellular mechanism for the degradation and recycling of material. Autophagy is involved in the removal of damaged organelles, misfolded as well as long-living proteins, lipids, cellular aggregations, parts of the nucleus and intracellular pathogens. These are engulfed by the phagophore membrane, which is matured and finally transported to lysosomes for degradation. Autophagy is a highly dynamic process and is stringently orchestrated by a large number of regulatory proteins. It preserves cellular homeostasis, for example, in situations of nutrient deprivation and acts as a defence against oxidative and proteotoxic stress. The current consensus is that autophagy communicates with almost all signalling pathways and organelles in the cell. Consequently, its failure has been linked with ageing and many diseases, like cancers, cardiovascular disorders and neurodegenerative diseases [142,143,144,145,146]. 

The impairment of autophagy has been proposed as one cause of the development and progression of AMD [39,142,147,148,149]. This is initially manifested by disturbed cargo management and impaired waste degeneration, as in the phagocytosis of POS. This leads to the formation of intracellular aggregates, which further elevate the proteolytic and OS in the RPE layer, leaving the tissue particularly severely exposed to oxidative tension. The link between autophagy and the NFE2L2 pathway is therefore interesting with respect to AMD.

As autophagy increases in cells in stressed conditions, this often coincides with the induction of the NFE2L2 pathway [150]. There are many recent reports of a positive correlation between these two processes. A number of inducers of NFE2L2 have been found to induce autophagy. For example, polyunsaturated fatty acids transiently increase the amounts of ROS in RPE cells and induce both NFE2L2 and the amount of protein aggregates containing SQSTM1/p62, a scaffold protein connected with autophagy [151]. Other compounds with comparable properties include sulforaphane, an isothiocyanate [152], dimethyl fumarate [153], dihydro-CDDO-trifluoroethyl amide, a triterpenoid derivative [154], fucoxanthin [155], RS9, a triterpenoid compound] [156] and fisetin, a flavonol compound [157]. Correspondingly, a deficiency of NFE2L2 results in reduced autophagic flux as shown by Dayalan Naidu and others [158]. 

SQSTM1/p62 is a protein which is involved in various signalling pathways since it contains several protein-protein interaction domains [159,160]. In autophagy, SQSTM1/p62 mediates the degradation of polyubiquitinated substrates, for example, protein aggregates, malfunctioning mitochondria and pathogens. It interacts directly with microtubule-associated protein 1 light chain 3 (MAPK1-LC3), an important protein in autophagosome formation. When the proteasome pathway, another route of removal of misfolded proteins, is inhibited, SQSTM1/p62 accumulates and this is accompanied by the co-localisation of ubiquitinated proteins [161]. SQSTM1/p62 increases the induction of NFE2L2, due to the association between itself and KEAP1, that is, there is an interaction between the ETGE-resembling motif and the KIR domain of KEAP1. The phosphorylation of Ser-349 of SQSTM1/p62 leads to greater affinity of SQSTM1/p62 to KEAP1 than the ^29^DLG motif of NFE2L2. This also causes SQSTM1/p62-aided autophagic degradation of KEAP1 and even greater activation of NFE2L2 due to its transfer to the nucleus. A positive feedback loop is thus formed for the activation of NFE2L2 via the autophagy-related SQSTM1/p62, in conditions of OS [162,163]. The overexpression of SQSTM1/p62 decreases the half-life of KEAP1, while its knockdown increases KEAP1’s half-life; similarly, a deficiency of the autophagy-related protein 7 (Atg7) increases the level of KEAP1 protein [122,164]. 

Thus, SQSTM1/p62 stabilises NFE2L2 and activates the transcription of NFE2L2-induced genes [165]. In addition, NFE2L2 can directly regulate the transcription of autophagy genes. Furthermore, SQSTM1/p62 has an ARE element in its promoter sequence, so it is inducible by NFE2L2 in situations of OS [123,166]. About 30 autophagy-related genes have ARE sequences in their promoter regions, of which NFE2L2 can regulate at least the following genes which are involved in all phases of autophagy: ULK1 (autophagy initiation), SQSTM1/p62 (cargo recognition), Atg7 and GABARAPL1 (autophagosome formation), Atg2B and Atg5 (elongation) and Atg4D (autolysosome clearance) [120,167]. In addition, and interestingly, in NFE2L2-deficient mice, the expressions of many of the autophagic markers mentioned above (and also that of MAPK1-LC3) have been found to be decreased [167]. 

The Ser/Thr kinase, AMP-activated protein kinase (AMPK), is the main sensor of cellular energy balance. It senses the levels of AMP and ATP and is activated when there are increases in the levels of AMP or ADP. Thus, AMPK couples NFE2L2 with autophagy. Firstly, it activates autophagy by inhibiting mammalian target of rapamycin complex 1 (mTORC1) [168]. AMPK phosphorylates the Ser-558 of NFE2L2 in its nuclear export domain and this in turn facilitates the accumulation of the transcription factor within the nucleus. Furthermore, AMPK inhibits the NFE2L2 degradation enhancer GSK-3β kinase by phosphorylation [92] (Section 2.4.1). This might be advantageous against the development of AMD [130] Energy depletion, especially when prolonged, might induce NFE2L2 through the action of AMPK, as well as stimulating autophagy. 

A proportion of cellular NFE2L2 is complexed with KEAP1 and the mitochondrial outer membrane histidine phosphatase 5 (PGAM5). O’Mealey et al. [169] have found that the depletion of either NFE2L2 or PGAM5 leads to the inhibition of mitochondrial retrograde trafficking. Mechanistically, this results from the degradation of Miro2 GTPase protein linking mitochondria to microtubuli, by the ubiquitination capacity of the released KEAP1-CUL3 complex and the proteasomal machinery, while the intact KEAP1-NFE2L2-PGAM5 complex prevents this degradation. This may be a possible explanation why the age-related decline of NFE2L2 exerts an impact on mitochondrial motility, a phenomenon which has been linked to degenerative diseases such as AMD.

### 2.6. NFE2L2 and Connections to RPE Functioning and AMD

The antioxidant signalling and the ARE response are generally supportive for the maintenance of the normal function of the RPE. The ageing RPE in mouse models has been observed to be vulnerable to OS due to impaired NRE2L2 antioxidant signalling. The ageing RPE has been found to express higher levels of NFE2L2 target genes such as HO-1 and NQO-1 as a response to increased OS. In old mice with a high ROS exposure, the NFE2L2 signalling has been found to be impaired and there is inadequate protection against OS. The knockdown of KEAP1 in the older mice restored the antioxidative capacity in RPE [170]. The enhancing of the functions of NFE2L2 could have therapeutic potential in the protection of RPE cells from OS-related damages and cell death and thus it could be beneficial against AMD, at least against the dry form of the disease. There is a direct, genetic link between AMD and this pathway, in that a single-nucleotide polymorphism in the *NFE2L2* gene (rs6726395), has been claimed to be associated with the risk for AMD [171]. 

Retinoid compounds, which are known to be important in the visual cycle and as signalling molecules in the retina, are potent chemopreventive and tumour-suppressive agents [115]. Their receptors include retinoic acid receptors (RARs) and retinoid X receptors (RXRs). For example, retinoid X receptor α (RXRα) binds to the Neh7 domain of NFE2L2 through its DNA binding domain (DBD) and thus it can block the transactivation of NFE2L2 [115,172]. A vitamin A metabolite, all-*trans* retinoic acid, can inhibit the NFE2L2 pathway. The gene expression of ARE-promoted genes has been found to increase in mice fed a vitamin A-deficient diet, which is in contrast to the situation in NFE2L2-null mice, where the vitamin A supply does not affect the ARE-driven gene expression [173]. The interactions between NFE2L2 and retinoic acid receptors could be of importance in determining how well cells can withstand OS in the retina [173], which has implications for the development of AMD. There might even be a connection between RARs and EMT, which will be discussed later (Section 3.5). It is interesting that all-*trans*-retinol, the “end-product” of the visual cycle, is transferred from POS to RPE to be recharged [13]. This might impact on the NFE2L2 activity in the RPE, that is, because of the persistent strong OS in RPE, all-*trans*-retinol might become oxidised to all-*trans* retinoic acid, which is an inhibitor of NFE2L2 [174]. This could therefore be a factor lowering the efficiency of NFE2L2-mediated anti-oxidative function within the RPE layer. 

With respect to AMD, sirtuin 1 expression has been found to protect the RPE against the functional changes as well as apoptosis and senescence, induced by OS and inflammation [175,176,177]. This can be partly be attributed to the activation of KEAP1-NFE2L2 pathway by sirtuins, as discussed in Section 2.3. 

Peroxisome proliferator-activated receptor gamma coactivator 1-α (PGC-1α, gene: *PPARGC1A*) regulates mitochondrial function and biogenesis and also controls the expression of many antioxidant genes. PGC-1α has been proposed to protect RPE cells against the degeneration induced by OS, by ensuring mitochondrial quality control and by regulating many of the processes involved in senescence [178]. NFE2L2 has been found to be co-activated by PGC-1α [179,180,181]. Furthermore, NFE2L2 might activate the transcription of *PPARGC1A* gene, which contains an ARE consensus sequence in its promoter region. It has still to be proved whether it is a functional connection. If it is true, it may possibly represent a regulatory loop [181]. A reduced expression of PGC-1α has recently been found to induce an AMD phenotype in mice and is associated with decreased autophagic flux [182]. In addition, PGC-1α expression has been claimed to be reduced in RPE cells gathered from AMD donors after the cells were exposed to H_2_O_2_ [183]. 

Retinal damage has been found in knock-out animal models of *NFE2L2* gene. A single knock-out animal model of *NFE2L2* gene has been shown to induce RPE degeneration, displayed as accumulation of lipofuscin and drusen-deposits, choroidal neovascularisation and increase of inflammatory proteins in RPE, as well as accumulation of autophagy-related bodies [40]. A double knock-out animal model of *NFE2L2* and *PPARGC1A* genes has been established. In the eye specimens of these animals, there is evidence of increased autolysosomes, augmented expression of autophagy markers (e.g. SQSTM1/p62 and MAPK1-LC3), damaged mitochondria, reduced proteosome activity manifested as an increase in the amounts of Ub-protein conjugates, RPE degeneration and finally age-related visual loss. These findings link mitochondrial damage to *NFE2L2* and *PPARGC1A* gene deficiencies and these may have an impact in autophagic and mitophagic clearance in the RPE and thus be involved in the pathogenesis of AMD [41]. 

NFE2L2 protects skin melanocytes against OS. Overexpression of NFE2L2 or downregulation of KEAP1 have been found to prevent the damage evoked by ROS of xenobiotic origin [184,185]. In skin melanocyte cells, NFE2L2 has been found to regulate melanogenesis. When cells were transduced with a viral cassette expressing NFE2L2, the synthesis of melanin was decreased. The mechanism behind this effect involved a modulation of PI3K/Akt signalling, since a PI3K inhibitor wortmannin could reverse the depigmentation. On the contrary, KEAP1 overexpression has been found to increase melanogenesis [186]. In addition, the *NFE2L2*-*PPARGC1A* double knock-out mice exhibited an increase in the numbers of melanosomes, that is, cells that contain the ROS eliminating compound, melanin. This could be a form of compensation to combat the otherwise reduced response to OS [41]. As RPE cells are melanin-rich and melanin protects cells against OS (Section 1.2), it would be of interest to examine whether NFE2L2 has similar actions in RPE as described above in skin cells and if this would be of any importance with regard to AMD.

## 3. Epithelial to Mesenchymal Transition (EMT)

### 3.1. General

The differentiation of epithelial cells to a mesenchymal phenotype, called the epithelial-to-mesenchymal transition (EMT), is essential in embryonic development, wound healing and stem cell differentiation and it contributes to pathological states of fibrosis. EMT has also been implicated in the initiation and progression of various cancers, when more malignant cell types appear [187]. Epithelial cells are normally joined laterally to form polarised, apico-basolateral layers with a three-dimensional organisation and usually they are non-motile. In contrast, mesenchymal cells do not generally have conjunctions with their neighbours. EMT is accordingly characterised by the loss of (1) cell-cell adhesion, (2) tight junctions, (3) desmosomes (*maculae adhaerentes*) and (4) apico-basal polarity in most cases. In addition, the switch from an epithelial morphology to spindle-like shapes and conversion to a mesenchymal phenotype is seen. The latter phenomenon includes remodelling of the extracellular matrix (ECM) and alterations to the cytoskeleton. These promote cell migration which can be intrusive as in the case of cancer metastasis. The process is characterised by actin rearrangement and formation of fibres. In addition, higher resistance to stress and apoptotic cell death are hallmarks of EMT [187,188,189]. The EMT switch can also occur in the opposite direction (MET) and there is also the possibility of partial transformations in both directions; these are especially required during the embryonic development of various tissues and cell types. These transformations highlight the plasticity of the epithelial cell type. Furthermore, an analogous process, the transformation of endothelial cells to a mesenchymal type, can occur in vascular cells [190]. EMT has been classified into three types, according to distinct biological dispositions. Type 1 is involved in the development of the organism, generating various cell types, which share a common mesenchymal phenotype. Type 2 is associated with wound healing, including scar formation, tissue regeneration and organ fibrosis for example, in kidney, liver, lung and eyes. This type of EMT is also associated with an inflammatory response. Type 3 occurs in neoplastic cells; it displays a clonal outcome and is important in both the development of tumours and the appearance of invasive metastatic cells [191]. In this review, we will concentrate on type 2 EMT and its association with AMD, which will be discussed in more detail in Section 3.5. Nonetheless, the mechanisms of EMT described in Section 3.2. do occur in all three types of EMT.

### 3.2. The Molecular Indications and Mechanisms of EMT

The EMT switch is controlled in the nucleus by a series of Zn-finger binding transcription regulators, including Snail family Zn-finger transcription factors SNAI1 and SNAI2 (SLUG). The other EMT factors include basic helix-loop-helix transcription factors, like Zn-finger E-box-binding 1 and 2 (ZEB1 and 2), lymphoid enhancer-binding factor 1 (LEF1), E protein factor E47 and TWIST factor [192,193,194]. Similarly, there are factors like NF-κB that maintain and promote the invasive EMT phenotype [195]. As an example of factors inducing MET, grainyhead-like 2 (GRHL-2) suppresses TWIST and ZEB1 [196]. In the next paragraphs, we will briefly review the molecular signs of EMT and the pathways, which launch it after the activation of the transcription factors, delivered by soluble and matrix-bound signals. Often these initiating signals are linked to OS. 

At the molecular level, EMT is characterised by the lowered expression of many proteins including E-cadherin, cytokeratins, laminin-1, desmoplakin, plakophilin, collagen type IV and tight-junction proteins, such as Zona occludens-1 (ZO-1), occludin and claudin. Conversely, the expressions of the following proteins are increased: N-cadherin, vimentin, fibronectin, metastasin (known also as FSP-1 or S100A4), fibroblast activator protein (FAP), mucin 1, α-smooth muscle actin (α-SMA), types I and III of collagen, connective tissue growth factor and androgen receptor-associated protein 55/Hic-5 (later Hic-5; a paxillin protein family member), some of which are used as mesenchymal markers [188,197,198]. For example, SNAI1 and 2, phosphorylated TWIST and SUMOylated ZEB2 transcription factors repress the expression of E-cadherin, while they simultaneously activate that of N-cadherin. Furthermore, they concurrently suppress many other epithelial proteins while activating many mesenchymal proteins. In addition, Snail transcription factors activate TWIST and ZEBs, further amplifying EMT [193,199]. Finally, there are losses of both apical-basal polarity and cell-cell adherence and the cells become spindle-shaped. Subsequently, these cells can invade the basal ECM and migrate to underlying tissues where they form a fibronectin matrix, which promotes fibrosis, as manifested in the type 2 of EMT. This is a process evident in the RPE as a symptom of AMD progression (Figure 1C,E) (Section 1.1).

The signalling pathways promoting EMT are normally aimed to the nucleus where they activate the transcription factors. We will begin with those routes directly connected to the OS (Figure 3). The signalling for EMT mediated by the transforming growth factor β (TGF-β) family is directly activated by ROS [200]. It is initiated by the binding of a TGF-β member to two cell membrane receptors, TGF-β RI and RII; this further activates various pathways. Of these, the SMAD (“Small mothers against decapentaplegic homologues”) pathway consists of SMAD 2/3, which is activated by phosphorylation and the cofactor, SMAD 4. These SMADs can activate EMT transcription factors directly. In contrast, SMAD 6/7 act as a suppressor of the pathway. In addition, SMAD 2/3-activates phosphoinositide-3-kinase (PI3K) which triggers the Akt pathway, which in turn inactivates GSK-3β, a suppressor of β-catenin. Catenin is involved in the regulation and gene transcription required for EMT. The TGF-β-activated Ras pathway stimulates extracellular signal-regulated kinases (ERK1/2). All of these pathways lead to the activation of several EMT-promoting transcription factors, like SNAI1, ZEB 1/2, LEF1 and TWIST [201]. Interestingly, the paxillin protein member, Hic-5, whose expression is increased in EMT, can negatively regulate SMAD3 [202] and thus suppress the TGF-β/EMT pathway. This may be an indication of the existence of counteracting or balancing pathways in the EMT process.

With respect to EMT, not only does TGF-β stimulate ROS production but there is also evidence that ROS can activate TGF-β in various cells [203]. TGF-β elevates ROS production via NADPH oxidase 4 (NOX4) and mitochondria. Conversely, ROS can induce the expression of TGF-β by oxidizing the latency-associated protein (LAP). TGF-β can be combined with LAP as a latent complex but the ROS-derived oxidation of the redox centre in LAP triggers a conformational change, releasing TGF-β. ROS can also activate matrix metalloproteinases (MMPs), which regulate the release of LAP. Moreover, TGF-β depletes cellular GSH, which increases the oxidative damage to macromolecules; all of these processes have significance with respect to fibrosis [200,204]. The increase in the ROS levels activates tumour suppressor protein p53, which regulates many cellular pathways. For example, p53 upregulates the transcription of TGF-β and its signalling [205]. Moreover, TGF-β has been found to be directly upregulated by the ROS produced by mitochondria [206]. 

Zhang and others [207] recently reported that elevated levels of ROS increase TGF-β1-induced EMT while inhibiting mitochondrial functions in lung cancer cells. In addition, they showed that resveratrol, a mitogen and an antioxidant inducing both NFE2L2 and PGC-1α, prevents EMT by preserving mitochondrial integrity [208]. In addition, mitochondria-derived ROS can upregulate distal-less homeobox-2 (Dlx-2) transcription factor, which can subsequently activate Snail factors [209]. These results suggest that one possibility to diminish EMT would be to preserve mitochondria; this could represent one possible way of shielding the RPE from EMT-derived fibrosis.

Furthermore, there are various other SMAD-independent signalling pathways for EMT and some of them are also ROS-inducible (Figure 3). Apart from the SMAD pathway, TGF-β RI can function directly as a transcription regulator, after being activated by TGF-β. The signal is mediated via partition defective 3 and 6 (Par3-Par6)-aPKC activation. In addition, Par3-Par6-aPKC can directly activate SNAI2, via the inactivation of Ras homologue gene family member A (RhoA), a repressor of this EMT transcription factor [201,210]. In growth factor/receptor tyrosine kinase (RTK) signalling, the presence of ROS allows the growth factors (e.g. EGF and platelet-derived growth factor; PDGF) to dock more easily to their tyrosine kinase receptors, which become dimerised with their Tyr residues being autophosphorylated. Furthermore, the signal is mediated via PI3K and Akt leading to the inactivation of GSK-3β. The subsequent activation of β-catenin triggers the stimulation of EMT. Growth factor-induced EMT-signalling can also be mediated via Ras, Src kinase (and further via p38 mitogen-activated protein kinase, MAPK) and the integrin-linked kinase (ILK) pathways, a cascade eventually triggering the expression of EMT-activating transcription factors [201,209]. 

Hypoxia, another type of stress associated with an oxygen imbalance producing ROS, causes the accumulation of hypoxia-induced factors (HIF). For example, the HIF-1, α subunit (HIF-1α), has been proposed to be regulated by various mechanisms, including NF-κB, ERK and PI3K/Akt signalling (Figure 3) [211,212]. HIF-1α activates EMT transcription factors by stabilizing β-catenin [213] or interacting with the intracellular domain (ICD) of NOTCH (a name derived from a *Drosophila* phenotype) receptor [214,215] (Figure 3). HIF-1α-NOTCH-ICD, which is translocated in the nucleus, acts as an EMT-transcription factor. Furthermore, hypoxia-induced NF-κB can act itself as an EMT inducer, by activating HIF-1α as well as TWIST and Snail factors [195,216,217]. Hypoxia can activate (“Wingless/Integrated”) WNT/β-catenin signalling for EMT. It has been proposed to activate the Axin degradosome complex, which inhibits GSK-3β, leading to the activation of β-catenin [218,219]. 

In addition, Hedgehog (Hh) signalling has been reported to be involved in hypoxia-induced EMT (Figure 3) [201,220]. In this pathway, the Hh ligand binds to patched homologue receptor (PTCH), which then evokes the release of the transmembrane protein Smoothened (Smo) from the membrane. Smo activates Gli family transcription factors, which in turn have many actions, for example, promoting the activity of SNAI1.

Additional EMT pathways that are less directly OS-dependent, reviewed by Cannito et al. [187] and Gonzales and Medici [201], are described here briefly. Irradiation-induced granulocyte-colony-stimulating factor (G-CSF) binds to its receptor, which activates Janus kinase/signal transducers and activators of transcription 3 (JAK/STAT3) pathway and subsequently STAT3 upregulates various EMT-related genes. Furthermore, the release of G-CSF is induced by the PI3K/Akt pathway [221]. In ECM signalling, type I collagen, the levels of which are augmented during EMT, induces the switch further by activating NF-κB and other transcription factors via the integrin and discoidin domain receptor (DDR) ligands. There are as well other EMT pathways included in matrix signalling, for example, ILK, PI3K and Paxillin-Src.

The importance of microRNAs (miRNAs) in the control of gene expression is known. Recently, miRNAs have been shown to contribute to EMT, either as inducers or inhibitors. They have been shown to affect TGF-β and WNT signalling, as well as directly participating in on the control of EMT transcription factors. These are involved in EMT-driven fibrosis and also in cancers [222,223]. By identifying those miRNAs possibly having an effect on the EMT-driven fibrosis in RPE, it might be possible to develop novel therapies not only against cancers but also to prevent AMD. It is evident that these kinds of therapies are still not on the horizon. At first, the specificity and safety of the response will need to be validated and then suitable delivery methods developed.

There are a vast number of EMT controlling pathways and as discussed above, many overlapping pathways exist. It is clear that there must be stringent and tissue-specific regulation of EMT in different cellular microenvironments and it is likely that there are multiple EMT/MET shuttles. Furthermore, there is evidence emerging of partial EMT/MET states; these are crucial in the type 1 of EMT in foetal development. This underlines the complexity of how these numerous EMT pathways interact with each other [201]. Since EMT is important in the progression of fibrosis in the eye, that is evident in AMD, while the prevention of EMT would be harmful in other occasions, this will be a major confront.

### 3.3. Connections Between EMT and NFE2L2 Signalling

After reviewing NFE2L2 signalling and EMT, we will now focus on the known connections between these two phenomena, especially concentrating on the question of whether it is OS that drives the processes (Figure 4). First, in cases of pulmonary fibrosis, NFE2L2 is intimately connected to EMT via its ability to directly suppress SNAI1 expression and thus EMT [224]. This can be achieved by administering either sulforaphane or rapamycin, which are both inducers of NFE2L2 [225,226]. The protective role of NFE2L2 against fibrosis might involve the downregulation of other proteins, for example, high-mobility group box 1, a transcription factor-like protein, which is abundant in epithelial and inflammatory cells [227]. It remains to be clarified whether NFE2L2 acts similarly in RPE to prevent EMT. In addition, indirect connections exist between these two processes but much is still to be revealed. NF-κB is one link between EMT and NFE2L2 signalling, as it activates the former, via HIF-1α and type I collagen signalling [201] (Section 3.2) and suppresses the latter [135,137] (Section 2.4.2). Conversely, KEAP1 itself can inhibit NF-κB via ubiquitination of IκB kinase β (IKKβ), an oncogenic kinase and a so-called fine tuner of NF-κB signalling [228]. These changes could counteract EMT and the myriad of other cellular processes associated with EMT. 

TGF-β1 represents a link between EMT and KEAP1-NFE2L2 signalling. In studies conducted in kidney cells TGF-β1 suppresses KEAP1-NFE2L2 antioxidant response pathway, by elevating the level of activating transcription factor 3 (ATF3). This in turn reduces the formation of GSH and induces fibrosis via EMT. Conversely, the knockdown of NFE2L2 induces TGF-β1-triggered EMT [229,230]. Surprisingly, in astrocytes, NFE2L2 seems to have the potential to induce ATF3, at least in stress conditions [231]. GSK-3β, a protein repressed by TGF-β/SMAD and growth factor pathways for EMT (Section 3.2), inhibits NFE2L2 by promoting its phosphorylation and preventing its nuclear translocation [232]. This connection is an evidence of the activation of NFE2L2 as a consequence of EMT. 

Another protein, SMAD ubiquitination regulatory factor 1 (SMURF1), an Ub E3 ligase and an important participant in several cellular pathways] [233], has been associated with both the TGF-β1-mediated EMT and KEAP1-NFE2L2 pathways. Firstly, SMURF1 activates SMAD-dependent EMT signalling by blocking SMAD7, the SMAD2/3 inhibitor [230] (Section 3.2). Furthermore, SMURF1 has been discovered to decrease the expression and nuclear translocation of NRF2L2, thus down-regulating the KEAP1-NFE2L2 pathway [234]. Casein kinase 2 interacting protein-1 (CKIP-1) has been reported to be a direct down-regulator of SMURF1. It thus activates NFE2L2 signalling and blocks EMT. CKIP-1 is inducible in OS, caused by oxygen-glucose deprivation/reoxygenation (i.e. conditions encountered in hypoxia) [234,235]. As far as we are aware, ROS themselves have not been reported to be connected to the activation of CKIP-1. Another recently reported link between EMT and NFE2L2 signalling is the upregulation of the HIF-1α-NOTCH axis seen after the knockdown of KEAP1 and overexpression of NFE2L2, these changes were reported to promote EMT-derived breast cancer cell migration [236]. This is one demonstration of the detrimental side of NFE2L2, as a promoter of cancer metastasis. These complex and at times contradictory associations between EMT and KEAP1-NFE2L2 need to be unravelled before it is possible determine whether they are relevant in the development of AMD.

### 3.4. EMT and Autophagy

The relationship between EMT and autophagy seems to be complex, at times contradictory and it is dependent on the cell type and the circumstances present in the cell. It has been shown that autophagy is involved in the activation of EMT in various tissues and cells, for example, malignant glioma cells, myofibroblasts and annulus fibrosus cells [237,238,239]. SQSTM/p62 is known to accumulate during growth-factor induced-EMT and has been shown to directly stabilise SMAD4 and TWIST in tumour-derived cell lines [240]. Recently, it was demonstrated that autophagy induces EMT in RPE cells. Stimulation of TGF-β2-induced EMT was seen when autophagy was induced by rapamycin. Correspondingly, inhibition of autophagy attenuated TGF-β2-induced EMT. In addition, RPE cells treated with TGF-β2 showed increased autophagic flux. These results suggest that the regulation of autophagy might be a potential way of reducing EMT-derived fibrosis in RPE [241]. Furthermore, hypoxia-induced HIF-1α (Section 3.2) controls autophagy by augmenting the expressions of Bcl-2/adenovirus E1B 19-kDa interacting protein 3 genes *BNIP3* and *BNIP3L*, which are also involved in mitophagy [242,243]. This connection adds hypoxia as another factor to be added into the EMT-autophagy network. Whereas inhibition of autophagy attenuates EMT, direct activation of autophagy in RPE cells increases EMT, perhaps because the activation of autophagy provides recycled ATP which can be utilized in the biosynthesis of profibrotic proteins [237].

Nonetheless, autophagy has been demonstrated to act as a repressor of EMT, depending on the cell type in question, for example, in hepatocytes, where transcription factor SNAI1 is degraded by autophagy [244]. Recently, autophagy has been found to be able to act as a brake on EMT in RPE cells. The mechanism behind this property was found to be the binding of SQSTM/p62 to TWIST, causing its degradation, although this result contradicts the findings of Bertrand et al. [240] described above. Moreover, the induction autophagy was found to promote the development of the epithelial phenotype. Correspondingly, EMT markers have been found to be increased in autophagy-deficient cells [245]. As seen, AMD is related to weakened autophagy and EMT occurs in advanced AMD. As autophagy and EMT seem to regulate each other and there are discrepant findings, further studies are needed to clarify their associations.

### 3.5. EMT and AMD 

In this final section, we review findings of the connections between EMT and AMD, since most of the literature is rather recent. The neovascular (wet) type of AMD results from choroidal neovascularization. Thus, it resembles the situation encountered in the neoplastic cancer cells and in metastasis, which has been described as type 3 of EMT. The end stage of AMD is associated with the formation of fibrous plaques and scars in the RPE and sub-retina. Fibrosis is usually connected to wound healing, in which angiogenesis is required for tissue repair at the wound location, in order to support the increased oxygen demand. It also promotes the interaction of cytokines and growth factors and the recruitment of inflammatory cells to the damaged site. There is also extensive modelling of the extracellular matrix [42,246,247]. Although anti-neovascularisation therapy has been found to be effective against wet AMD, the ineffectiveness of this treatment in some patients is due to subretinal fibrosis (Figure 1C,E), possibly induced by EMT. EMT has been implicated in other intraocular fibrotic disorders, like proliferative vitreoretinopathy [207,248]. Galectin-1, a galactoside-binding lectin protein, is one of the recently found proteins thought to be involved in both wet AMD and EMT. It induces VEGF and TGF-β signalling pathways, facilitating neovascularisation and fibrosis in RPE cells. One mechanism behind the fibroplastic morphology of EMT is the galectin-1-driven promotion of TGF-β/SMAD-signalling [249]. Very recently, it was reported that the silencing of PCG-1α protein, the master regulator of mitochondrial function and genesis (Section 2.6), was able to induce retinal degeneration in mice by inducing EMT in RPE, as well as disturbing energy sensor activity, mitochondrial function and autophagic clearance [250]. This suggests that PGC-1α maintains the autophagy-mediated repression of EMT, as well as supporting the oxidative metabolism of the RPE.

EMT might also be of importance in the more common, dry form of AMD [251]. Crystallin proteins provide the transparency and refractivity in the lens of the eye. Participation of βA3/A1-crystallin in retinal signalling has been shown recently. Firstly, a deficiency of this crystallin leads to a failure in the lysosome-mediated clearance in RPE cells, as it interacts with V-ATPase, a proton pump needed in the acidification of lysosomes. Secondly, if there is a shortage of βA3/A1-crystalline, RPE cells tend to avoid the stress by activating EMT. The disruption to lysosomal clearance can be of importance in the development of dry AMD. The loss of the *CRYBA1* gene, coding for βA3/A1-crystallin, leads to EMT in RPE cells, as manifested by the reduction in the amount of E-cadherin and increases in expression of Snail factors and vimentin, as well as increased migration of the cells. These changes are reminiscent of those encountered in wound healing, as well as the morphological changes observed in type 2 EMT [251]. In addition, the loss of *CRYBA1* in mice leads to failures in autophagic flux and impaired clearance in the RPE in vivo [252]. On the contrary, and intriguingly, Ishikawa and others [253] have reported that αB-crystallin knockout decreased EMT, with overexpression having an opposite effect. The mechanism of action of αB-crystallin is that it acts as a molecular chaperone for SMAD4. This suggests that different family members of crystallins might differentially regulate EMT.

Hypoxia is known to be associated with EMT in AMD, as the vascular development in the retina is dependent on HIF-1α [39]. In addition, hypoxia and ROS activate NF-κB, which directly controls HIF-1α. Consequently, in hypoxic situations, inflammation is stimulated, RPE-associated deposits are accumulated and retinal neovascularisation occurs, all hallmarks of the progression of AMD [216]. HIF-1α is also a recognized activator of autophagy and mitophagy; it is released in response to hypoxia to promote cell survival [243,254]. The activation of autophagy by hypoxia occurs independently of HIF-1α [255]. When the autophagy rate declines in aged cells, the removal of hypoxia- and ROS-derived deposits in the RPE is weakened and AMD progression can occur. Nevertheless, one must view the suppression of HIF-1α as a therapy for AMD with caution, since this factor has beneficial effects in regulating the clearance of damaged proteins.

The upregulation of keratin 8 (KRT8) and its phosphorylated form p-KRT8 are connected with increased autophagy. Conversely, during OS in RPE cells, KRT8 is downregulated leading to the inhibition of autophagy [256]. The increase in the amount of p-KRT8 has been connected with increased EMT. Initially, ROS could induce mitogen-activated protein kinase kinase 1 (MAP2K1) and further MAPK1, which would increase the p-KRT8 induction of EMT and ultimately promote RPE degeneration. Inhibition of autophagy induces the cellular dedifferentiation—a phenomenon characterised by a reduction in the levels of RPE-related proteins and cellular hypertrophy, induction of EMT and degeneration of RPE during OS. Thus, enhancement of autophagy and increased expression of KTR8 with inhibition of p-KRT8 could inhibit the EMT induced by OS [256]. 

An attenuation of fibrosis in RPE cells by suppression of EMT has been reported by Kobayashi et al. [257] and Kimura and others [258]. CCG-143 (a fluoro-containing benzamide compound), an inhibitor of myocardin-related transcription factor A (MRTF-A) has been found to suppress RPE migration in cell culture experiments. The inhibitor reduces the expressions of type III collagen and α-SMA, a connective tissue growth factor. Even an intravitreal injection of CCG-143 in mice was successful against fibrosis [257]. Palovarotene (compound R667), an RAR-γ agonist, has been claimed to suppress EMT and fibrosis-related processes. Cell movement, as well as the reduced expressions of α-SMA and fibronectin were observed. In vivo fibrosis in mice was found to be inhibited by treatment with an RAR-γ agonist [258]. In that report, it was interesting that manipulating the biochemistry of a receptor involved in the metabolism of the components of the visual cycle could exert a beneficial effect in EMT in RPE cells. It is moreover worth noting that NFE2L2 is inhibited by a member of the same receptor family, RAR-α and by all-*trans*-retinoic acid, a ligand of the RARs (Section 2.6).

Cigarette smoke is known to induce EMT along with ROS production, as revealed by the increase of N-cadherin and the decrease in the levels of epithelial markers. In endometrial adenocarcinoma cells, these effects were attributed to the upregulation of SNAI1 and 2 transcription factors [259]. Smoke activates both the p38 MAPK and ERK1/2 pathways (Section 3.2). This could be of importance in type 2 EMT with regard to AMD, as smoking is one of the most important risk factors for this disease. Recently, blue light-emitting diode (LED) light irradiation has been shown to induce ROS production and increase EMT, in addition to promoting cell proliferation and DNA damage in colorectal cancer cells [260]. 

A reduction of mitochondrial activity has been postulated to be involved in AMD [261,262,263]. It has been shown that mtDNA damage induces EMT-like reprogramming, leading to a fibroplastic morphology [264]. The knockdown of mitochondrial Tu translation elongation factor (TUFM) induced EMT and reduced the efficiency of the respiratory chain [265]. Scavenging of mitochondrial ROS was reported to evoke a reversal of EMT in cervical cancer cells [266]. All of these findings suggest that EMT, which is linked with reduced mitochondrial activity, if likewise manifested as is thought to occur in type 2 EMT, might be at least partly related with the progression of AMD.

It must be emphasized here, that we have discussed an array of OS-dependent EMT pathways (Figure 3), many of which have been initially associated with various forms of cancer, classified as either type 3 EMT or fibrosis (type 2 EMT) in other organs like kidneys and not necessarily in the eye or the RPE. Nonetheless, most of these pathways have been thought to be involved in tissue repair, wound healing and fibrosis, that is, type 2 of EMT [223,267]. It remains to be clarified whether all of these EMT signalling routes are valid in the fibrosis encountered in RPE and thus would be relevant to AMD. These issues may well be topics for future research.

The question remains whether the EMT phenotype is more important in the dry form of AMD or is it an intermediate stage in the development of exudative AMD. When considering the role of EMT in AMD in general, it is important to remember that RPE cells are post-mitotic and non-dividing, whereas cancer cells are proliferative. With regard to EMT, different signalling mechanisms might act in the distinct types of cells and in their different stages and therefore it is clear that much more research is needed in this promising area of research. Furthermore, it might be worthwhile to consider whether the complete destruction of the RPE and its neighbouring tissues which occurs in the very advanced, “terminal” phase of AMD, could be at least partly due to the increased detachment and then loosening and elevated motility of the cells, caused by EMT, in addition to the fibrosis-evoked damage occurring during this transition.

## 4. Conclusions

According to current knowledge, OS is responsible for much of the damage connected to the initiation and progression of AMD and thus, if there were a means of controlling this excessive OS, it could provide a novel approach to treat this disease. Here, we have discussed the role NFE2L2 signalling-associated pathways in the detection, signalling and mediation of the cellular defence response against oxidation. The EMT transition, our other topic, is not only a feature of foetal development and cancer expansion but in fibrosis, it is usually associated with excessive wound healing processes and these have been linked to the damage to the RPE layer that occurs during AMD progression, especially of its wet form.

The management of the NFE2L2 pathway may provide a potential and novel treatment of AMD, by lowering the damage induced by OS. For example, certain non-electrophilic inducers, which directly block the DG-DLG binding between KEAP1 and NFE2L2, could perhaps achieve this target. These compounds would not cause electrophilic attack of the cell, in contrast to many electrophilic compounds, which bind to the Cys residues of KEAP1 (Figure 2B) [48]. Furthermore, an induced version of NFE2L2 has been shown to increase the expression of ARE-connected antioxidant proteins [268]. Similarly, the control of oxidation-induced EMT, via the various pathways related to OS, could be one way to avoid or diminish the damage to the RPE, perhaps not only in the therapy of the wet type AMD but possibly also in combatting the much more common dry AMD. The signal for EMT is transmitted via a plethora of pathways, many of them directly enhanced by OS and linked to the NFE2L2 signalling. There are certainly many potential targets to delay or even prevent the progression of AMD. In addition, the linking of autophagy to both of NFE2L2 antioxidant signalling and EMT adds an interesting dimension to the biochemical pathology of AMD, as reduced autophagy has been thought to be a possible cause for the development of this disease. 

Given the complexity and diversified mechanisms associated with the interaction between NFE2L2, EMT and OS, more research will be needed before a clear message emerges from ongoing signalling studies that will offer clues for the therapeutic application to models of AMD.

## Figures and Tables

**Figure 1 ijms-20-05800-f001:**
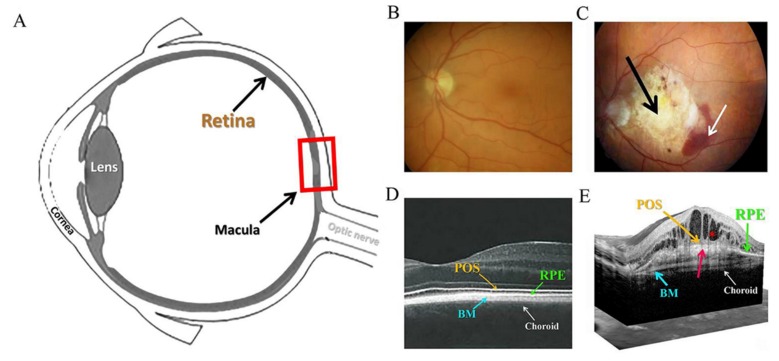
(**A**) Structure of the human eye, showing the retina and macula. (**B**) Fundus photograph of a healthy eye. (**C**) Fundus photograph of a fibrotic (black arrow) and treatment-tolerant wet age-related macular degeneration (AMD) eye. White arrow indicates the haemorrhage associated with active choroidal neovascularization. (**D**) Optical coherent tomography (OCT) image of healthy retina with the arrows pointing to the tissue segments. (**E**) Three-dimensional OCT image of fibrotic AMD retina. Fibrotic region is marked by a red arrow and apparent intraretinal oedema by a red asterisk. Abbreviations are included in the list of abbreviations.

**Figure 2 ijms-20-05800-f002:**
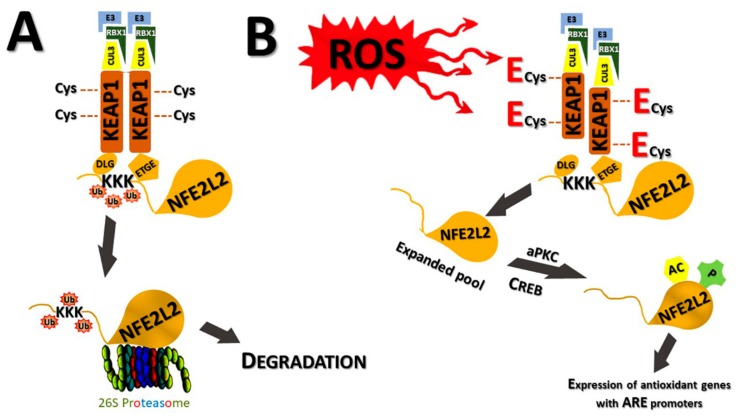
The canonical KEAP1/NFE2L2 antioxidant signalling pathway, induced by electrophilic ligands. (**A**) In basal conditions, KEAP1 dimer and NFE2L2 are bound together, with the DLG and ETGE amino acid domains of NFE2L2. This sustains the constant ubiquitination of NFE2L2 and its resulting turnover by proteasomal degradation. (**B**) In OS, Cys-residues of KEAP1 bind electrophilic ligands (E), leading to a loosening of the DLG-binding, a subsequent cessation of ubiquitination and finally the detachment of NFE2L2 from the complex. After its transfer to the nucleus, NFE2L2 acts as a transcription factor for ARE-dependent antioxidant genes. Abbreviations: CUL3, adaptor component Cullin 3; DLG, Asp-Lys-Gly motif; E3, E3 ubiquitin ligase; ETGE, Glu-Thr-Gly-Glu motif; KKK, lysine-rich domain; RBX1, RING box protein 1. Other abbreviations are explained in the list of abbreviations.

**Figure 3 ijms-20-05800-f003:**
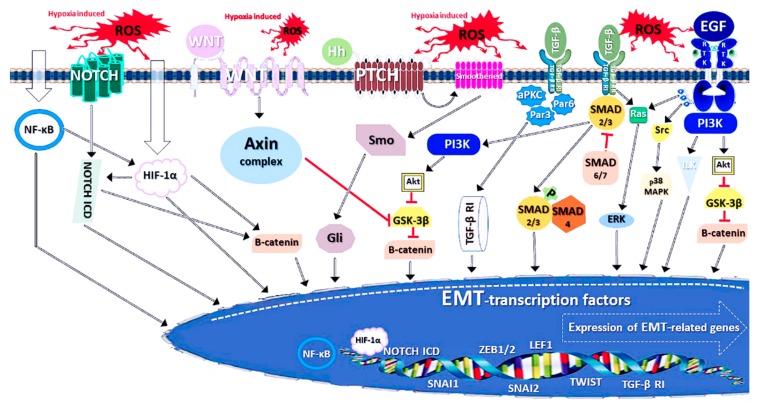
The known OS-dependent mechanisms of epithelial-to-mesenchymal transition (EMT), represented in simplified terms. These include signalling mediated by transforming growth factor β (TGF-β), growth factor/RTK (here epidermal growth factor; EGF), Hedgehog (Hh)-Smoothened receptor, WNT (“Wingless/Integrated) receptor, NF-κB, HIF-1α and NOTCH receptor. Here, the arrows pointing from the signalling routes to the EMT transcription factors (not an exhaustive collection) to the DNA in the nucleus are grouped. More specific information about the activations of certain transcription factors can be found in reviews in the literature. The mechanisms of HIF-1α activation by hypoxia-induced reactive oxygen species (ROS) in the far left of the figure, are still somewhat tentative and beyond the scope of this review. Abbreviations: ERK, extracellular signal-regulated kinase; Gli, Gli family transcription factor; Hh, hedgehog; ICD, intracellular domain; ILK, integrin-linked kinase; LEF, lymphoid enhancer-binding factor; p38 MAPK, p38 mitogen-activated protein kinase; Par, partition defective; PTCH, patched homologue receptor; RTK, growth factor/receptor tyrosine kinase; Smo, smoothened (intracellular); SNAI, Snail family Zn-finger transcription factor; ZEB, Zeb family Zn-finger transcription factor. Other abbreviations are included in the list of abbreviations.

**Figure 4 ijms-20-05800-f004:**
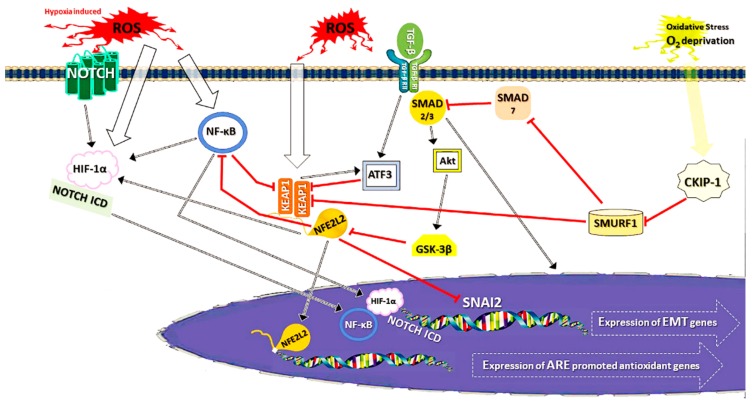
The currently known links between NFE2L2 signalling and epithelial-to-mesenchymal transition (EMT). These links are explained in detail in the text. Abbreviations: ATF3, activating transcription factor 3; CKIP-1, casein kinase 2 interacting protein-1; SNAI2, Snail family Zn-finger transcription factor 2. Other abbreviations are detailed in the list of abbreviations.

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
