# Peer review of "The Regulation of NFE2L2 (NRF2) Signalling and Epithelial-to-Mesenchymal Transition in Age-Related Macular Degeneration Pathology"

_ijms, 2019, doi:10.3390/ijms20225800_

Round 1

Reviewer 1 Report

The present review collects and presents interesting biochemical data on the possible involvement of two signaling pathways in the progression of AMD unfortunately the link between molecular data and the pathology is indirect there are suggestions but it has to be pointed out clearly that there are no available data on possible treatment (even in animal models). The title is too ambitious! For ex. in the reported literature there are evidence of a connection between noxious stimuli and a decrease in the level of NFE2L2 but still the message remains unclear, the link between NFE2L2, NF-kB and neuro-inflammation is important but again it is matter od speculation. The connection between NFE2L2 and EMT in the development of AMD remains matter of speculation. In general the paper contains lot of information but it is not well organized and the paper itself is too long and the core message remains unclear. The paper is potentially interesting but it needs to be presented in a more convincing way.

Reviewer 2 Report

The review 'The intimate and critical link between oxidative 2 stress in age-related macular degeneration (AMD) 3 and regulation of NFE2L2 (NRF2) signalling and 4 epithelial-to-mesenchymal transition (EMT)' by Hyttinen and colleagues is a very comprehensive study into the role of NFE2L2 and EMT in RPE cells and ultimately AMD. The authors have done a great job in compiling this very thorough review, trying to explain every possible oxidative stress response. Especially the figures help to understand the mechanisms.

There are however a few issues with the language style, the list of abbreviations and general format of the text. Once these issues have been addressed I would be very happy to see this review published.

Throughout the text grammatical errors can be found. For example sometimes words are missing or the sentences are too convoluted so that the meaning is obscured. Some of the abbreviations found in the text are not listed in the list of abbreviations. Examples include SOD1 (line 79). Please add all abbreviations to the LoA. I feel that a reference should be placed at the end of the sentence ending in line 101. References 17-18 are fine but maybe a newer analysis should be included. For example: Wong WL, Su X, Li X, Cheung CMG, Klein R, Cheng CY, et al. Global prevalence of age-related macular degeneration and disease burden projection for 2020 and 2040: a systematic review and meta-analysis. Lancet Glob Health. 2014;2:e106–e116. doi: 10.1016/S2214-109X(13)70145-1 Sometimes e.g. and i.e. are italicised and sometimes not. For example line 111 and line 533. The sentences ending in lines 153 & 155 should have references. Even though the risk factors of AMD might be common knowledge I feel that a recent review should be cited here.  In line 219, the heading should have the full name written out - The Kelch-like ECH-associated protein 1 (KEAP1)-nuclear factor erythroid 2-related factor 2 (NFE2L2) pathway In lines 230/231 NFE2L2 should be abbreviated. Line 267: AIB1 should be written out as amplified in breast cancer 1 the first time it is mentioned Line 295: AREs have been abbreviated in line 262 already There are some problems with the reference format throughout the text. I encourage the use of a reference manager. See for example line 353 [1079], line 370 [114] and line 680 as well as line 805. Line 419: There should be a reference here. I found it confusing that Keap1 is sometimes capitalised and sometimes not. Did you mean protein vs gene? Line 476: nuclear factor kappa-light chain-enhancers of activated B cells should be written out here as first time used Line 876: ECM has been abbreviated before line 889 onwards: format of text

Round 2

Reviewer 1 Report

The paper has been improved and, in the present form, it conveys a clear and important message. My only concern is about how the role of pigment epitelium is presented in AMD in line 61 the authors used the word “primarily” refers to pigment epitelium , I would better say that pigment epitelium plays a relevant role in maintaining photoreceptors function but it is not the only actor in degenerative processes. Oxidative stress might also acts directly on photoreceptors. The same message can be given with less emphasis

Author Response

Responses to Reviewer 1

 The opening part of the paragraph beginning from line 65 has been modified. Following notes are closely considered:

The authors used the word “primarily” refers to pigment epitelium.

Word “primary” has been removed.

Pigment epitelium plays a relevant role in maintaining photoreceptors function but it is not the only actor in degenerative processes.

Stated in line 67: Not only the RPE, but also the photoreceptor outer segments (POS) and the choroid, are degraded in AMD.

Oxidative stress might also acts directly on photoreceptors.

Text beginning from line 103 indicates this. ROS-derived damaged material originating from photoreceptor outer segments, cleansed by RPE, is of relevance in the macular degeneration.